# Corporate Sustainability Development Strategy and Corporate Environmental Governance—The Moderating Role of Corporate Environmental Investments

**DOI:** 10.3390/ijerph20054528

**Published:** 2023-03-03

**Authors:** Xiangyuan Ao, Tze San Ong, Boon Heng Teh

**Affiliations:** 1School of Business and Economics, Universiti Putra Malaysia, Serdang 43400, Malaysia; 2Department of Business Administration, Daffodil International University, Dhaka 1341, Bangladesh; 3Faculty of Management, Multimedia University Malaysia, Cyberjaya 63100, Malaysia

**Keywords:** corporate green behaviour, corporate environmental strategy, corporate environmental investment, sustainable development

## Abstract

Environmental degradation and ecological devastation have become widespread global concerns in recent years as a result of the expansion of the international economy. China’s rapid economic development has been accompanied by a sloppy economic growth model that has damaged the local ecological environment. The Chinese government intends to improve the ecological environment by the end of 2020 in an effort to direct and improve these environmental issues. The strictest environmental laws became effective in 2015. In light of this, this research uses panel data analysis to examine the environmental strategy and environmental governance of Chinese corporations. This article analyses 14,512 samples of listed mainland Chinese enterprises from 2015 to 2020. This research investigates the connection between Corporate Sustainability Development Strategy and Corporate Environmental Governance, as well as the moderating effect of Corporate Environmental Investments.

## 1. Introduction

In recent years, environmental degradation and ecological devastation have become common global issues due to the rise of the global economy. Corporate sustainability, environmental reduction, and overall economic growth may not “go hand in hand” [1,2]. Since 2010, China’s economy has become the second largest in the world [3,4]. Its rapid economic rise appears to be correlated with inefficient energy consumption and environmental degradation [5]. Liu et al. [6] noted that a substantial number of Chinese firms have recognised the significance of environmental management and have attempted to enhance their ecological performance through various approaches. To reduce environmental degradation, businesses should invest more in environmental protection and sustainable economic development, and corporate environmental investment can strengthen corporate environmental governance [7]. This is an effective microlevel strategy for reducing environmental problems caused by firms’ excessive resource extraction and energy consumption [8]. In reality, when confronted with actual environmental issues, Chinese businesses must find a balance between speed and quality in their economic development model: quick economic expansion while compromising on excessive resource use and environmental destruction. Corporations are major polluters of the environment, and their manufacturing and operating methods have a substantial effect on national environmental protection and energy conservation [7].

Environmental investment, on the one hand, is likely to affect firms’ operations as budgets are constrained [9,10]; on the other hand, environmental investment reduces costs through advanced technologies [11,12] and builds up corporate reputations with sustainable operations [13], thereby creating an invaluable asset for firms [14] and enhancing firm performance [15]. Consequently, research on corporate environmental investment is crucial for modifying not just China’s environmental governance but also worldwide environmental governance. The crucial role that corporations play in tackling environmental issues has, however, received limited scholarly attention. Braam et al. [16] propose that corporate environmental performance is becoming increasingly accountable for the environment and encourages businesses to increase their environmental accountability. Subsequent research supports and expands upon the hypotheses. Economic growth is positively affected by corporate environmental governance, according to Liang and Liu [17]. From a cross-country and single-country viewpoint, [18,19] found that suitable corporate environmental investment strategies minimise costs and risks, hence promoting the sustainable development of businesses.

In fact, there is no specific research on how the environmental investment of Chinese-listed firms influences the connection between the corporate sustainable development plan and the environmental governance outcomes of these corporations [7,20]. This study aims to examine the consequences of corporate sustainability plans and corporate environmental governance, with a particular emphasis on the moderating effect of corporate environmental investments. This study’s findings will assist more Chinese businesses in obtaining empirical evidence to play a role in the Sustainable Development Goals and corporate environmental governance. In addition, it gives evidence for the Chinese government to strengthen its economic development and environmental policies.

Consequently, the goal of this study is to examine the moderating effect of environmental investments on the link between corporate sustainability and environmental governance in China-listed enterprises. This part focuses on introducing the study’s research context and aims. The remainder of the research is structured as follows: The Section 2 involves a literature review and hypothesis formulation. The Section 3 outlines the research methods used. The results of the data analysis are reported in Section 4. The Section 5 contains the discussion and conclusions. This study’s limitations and implications are discussed in its conclusion.

## 2. Literature Review and Hypothesis Development

This section defines environmental uncertainty, carbon responsibility, green innovation, environmental governance, and environmental investment. From theoretical and empirical perspectives, the relationship between these sustainable development strategies (environmental uncertainty, corporate carbon responsibility, and green innovation) and environmental governance will be defined. In addition, the role of environmental investment is elaborated upon in this paper. This study concludes with arguments and hypotheses based on the literature and proposed measures.

### 2.1. Sustainable Development Strategy and Environmental Governance 

According to the environmental economics theory [21,22], the expansion of an economy is directly proportionate to the growth of its ecological environment, and achieving equilibrium and coordination between the environment and the economy is essential. Environmental economics provides a significant theoretical foundation for the subjects of environmental economics and environmental governance [8]. Environmental economics emphasises the interaction between economic development and the environment, as well as the coordination of the relationship between humans and nature, all while serving the expanding material demands of the market. Environmental economics always considers sustainable development to be the principle upon which enterprise growth is based [23,24]. Therefore, corporations must undertake sustainable development in tandem with their own growth. Corporate sustainable development strategy must apply relevant theories of environmental economics, such as environmental uncertainty, corporate carbon responsibility, and green innovation [8,9], make decisions that are beneficial to both corporations and the environment, and achieve a win-win situation between corporations and environmental protection [8,9,25]. Based on the preceding logic, this study concludes that the sustainable development strategy of businesses has some effect on the outcomes of environmental governance.

A resource-based theory can explain the relationship between a deteriorating environment and a developing economy. The year 1984 saw the introduction of Wernerfelt’s resource-based theory, which posits that organisations own distinct tangible and intangible resources that may be converted into distinctive capabilities. The Resource-based View (RBV) theory is significant in the field of strategy because it has the ability to explain sustained competitive advantage, which is the process of generating anomalous long-term returns to shareholders [26,27]. Ref. [28] concur with the beneficial results of corporate environmental investment and claim that investment in new technology leads to reduced energy consumption and, consequently, fewer pollutant emissions. Consistent with resource-based theory, sustainable development techniques, and environmental governance outcomes are seen as intangible assets and competitive advantages of businesses.

Moreover, in an unstable global capital market [29] and a volatile investment environment [30], businesses that apply a responsible approach to environmental concerns would earn greater stock returns [30,31,32,33]. Nonetheless, a number of researches have revealed that businesses incur higher compliance costs for environmental protection and their profitability is worse than predicted [34,35]. Reduced investment capacity was a result of rising variations in future cash flows [36,37]. Due to the increased emphasis on green recovery and environmental responsibility in the latter phases of environmental turbulence, corporations with more robust sustainability strategies are likely to display superior crisis management and more resilience [38]. This is due to the rising emphasis on green recovery and environmental stewardship in the latter stages of environmental instability [39]. Businesses with effective environmental governance can provide the market with more positive signals, indicating enhanced environmental adaptation and resource utilisation [40]. This study, therefore, proposes the first hypothesis:

**H1:** *There is a significant relationship between environmental uncertainty and corporate environmental governance*.

In accordance with the Paris Agreement of 2016, which established the objective of global carbon neutrality, China suggested in its 14th Five-Year Plan to accelerate green growth and build an action plan for peaking carbon emissions by 2030. With the steady disclosure of environmental concerns, the public’s consciousness of environmental protection is continually awoken, and an increasing number of businesses are pursuing sustainable development [41]. Carbon responsibility is considered a new level of corporate environmental responsibility [42]. Corporate entities are major producers of carbon emissions and consumers of energy [43], with global greenhouse gas (GHG) emissions as the primary monitoring and measurement of carbon emissions [44]; consequently, their manufacturing and operation methods have a substantial impact on environmental governance. In addition, this research proposes the second hypothesis:

**H2:** *There is a significant relationship between corporate carbon responsibility and environmental governance*.

“Going Green” is a corporate program that primarily addresses environmental concerns. Approaches to gaining green competencies and environmental governance have been a topic of discussion in the field of management science [45]. As a crucial means for businesses to gain green capabilities, green innovation has become an integral element of the strategic policies and tactical strategies of a great number of organisations [2,46]. Long-term green innovation in business can assist in improving energy efficiency, encouraging recycling, reducing pollution, and accomplishing other environmental governance goals [7]. This study, therefore, proposes the third hypothesis:

**H3:** *There is a positive significant relationship between corporate green innovation and environmental governance*.

### 2.2. The Moderating Effect of Environmental Investment

According to the Environmental Kuznets Curve (EKC) effect, economic expansion does not lead to a continual worsening of the environment for major economies such as China. When the economy reaches a particular level of growth, environmental damage is reduced [47,48]. Consequently, this influence fluctuates when there is environmental uncertainty and a conflict between carbon emissions from commercial operations and China’s ‘net zero’ carbon emissions aim, resulting in unsatisfactory environmental governance outcomes. In addition, environmental investment is commonly regarded as the expenditures made by companies to invest in environmental activities using green capital mobilised by the government and industry in order to achieve long-term social and economic development by harmonising and aligning economic, environmental, and social benefits [49,50]. As a result of environmental unpredictability, corporate environmental investments may fluctuate and eventually influence environmental governance results. This study proposes the following:

**H4:** *There is a moderating effect of environmental investment between environmental uncertainty and environmental governance*.

The fundamental objective of corporate carbon responsibility is to achieve sustainable business in the near future, which requires the company to operate in a sustainable natural environment. This clarifies the two-way objective link between the company based on carbon emissions and the natural environment [42,51]. Consequently, the emphasis of corporate carbon responsibility should change from the output to the cost to the value that the environment provides to the organisation, as well as from the former concentration on results and efficiency to a focus on the long term [52]. Therefore, corporate environmental investments will influence the acceptance of carbon responsibility. The study provides a fifth possibility in this area.

**H5:** *There is a moderating effect of environmental investment between carbon responsibility and environmental governance*. 

The sustainability development strategy is a long-term strategy, and when the external environment is challenging, corporations tend to choose a conservative investment strategy, reducing funds for innovative investments and maintaining a high free cash flow [53] in order to deal with market shocks and fierce market competition and to alleviate the pressure for survival. The stronger the tension between corporate investment and plans for sustainable growth, the less accurately corporate management analyses the benefits of green innovation investment projects, preferring to defer innovation investments or reduce capital expenditures [54]. Thus, environmental investment promotes green innovation within corporations, which in turn influences environmental governance within corporations. This study proposes the following hypothesis:

**H6:** *There is a moderating effect of environmental investment between green innovation and environmental governance*. 

Figure 1 illustrates the research framework for this paper, which includes both direct and moderate relationships.

## 3. Methodology

### 3.1. Research Samples and Data Sources

This study gathers the financial data of domestic A-share listed businesses from 2016 to 2020 from the China Stock Market and Accounting Research database and sorts and analyses the data using the data processing software Stata15 and Excel (Microsoft, Redmond, WA, USA). A total of 14,512 samples were obtained after excluding ST-listed businesses, financial, and insurance-listed companies, and samples with missing data on corporate carbon responsibility, environmental investment, green innovation, and environmental governance factors. In addition, this study winsorized the top and bottom 1% quantiles of continuous variables in order to reduce the gap between extreme values and empirical data.

### 3.2. Definition of Main Variables

#### 3.2.1. Explained Variable (Y)

Environmental governance was used as an explanatory variable in this study. This variable’s measuring elements were derived from current advancements in environmental governance [55] and the corporate environmental governance scoring system based on the HEXUN database. Using Liu et al. ’s [40] corporate environmental governance scheme as a foundation, this study selects five indicators that are closely related to environmental governance in areas such as environmental protection concepts, implementation, environmental management status, pollution control, and the use of clean energy.

#### 3.2.2. Explanatory Variable (X)

Environmental uncertainty, green innovation, and corporate carbon responsibility as three explanatory variables in this study. 

Environmental uncertainty has been characterized in terms of dynamism. Dynamism refers to the environmental instability that makes it difficult to predict changes and affects the volatility that a business unit faces. Typically, the volatility of industry sales and income is used to proxy dynamism [56]. In addition, the company’s operating income data from the past five years is used to calculate the standard deviation of abnormal sales income in the past five years to measure the fluctuation of its income. Then, it is adjusted in consideration of industry standards, and the industry-adjusted value is calculated as the environmental uncertainty. See Equation (1) for details.
Sale = φ0 + φ1Year + ε(1)

Among them, the Sale is the operating income, and the Year is the annual variable.

Green innovation. There is a general inconsistency in measuring green innovation in previous studies. Some studies use R&D expenditures to measure [57,58], but this may overestimate innovation [59]. In addition, there is less likelihood for firms to report their environmental R&D spending, so the data on spending are not available for all sample firms [26,36]. Thus, this study uses the number of green patents disclosed by enterprises each year. 

Corporate carbon responsibility (CR) is mainly to describe the performance of corporate on carbon emission. Based on the recent literature [48,60], this study uses the logarithm of corporate Greenhouse gases (GHG) emissions reductions to measure corporate carbon responsibility. The data used in the carbon emission calculation is basically from various statistical yearbooks published by the National Bureau of Statistics of China. 

#### 3.2.3. Moderator Variable

Environmental Investment (EI) is the moderator variable in this study, as evaluated by the sum of current occurrence amounts on ongoing environmental protection fees (in 10,000 CNY, equivalent to about 1508 USD at the 2018 exchange rate) divided by total profits [7,50]. A company’s strategy for establishing and retaining legitimacy in the environmental sector, which impacts corporate performance, is assessed by green investment. To explore a specific response, they concentrated on environmental-related green investment and investors, such as managers, regulators, chief procurements, producers, and company managers. Consequently, the present study sought to quantify and apply the results of two prior studies. As stated previously, the ratio of total environmental protection investment to corporate capital stock is used to reflect the degree of environmental protection spending by corporations in this study. Total assets at the beginning of the year plus total assets at the end of the year equal the capital stock. EI was carefully collected from the annual reports and corporate social responsibility reports of publicly traded companies.

#### 3.2.4. Control Variables

According to current research, this study adds a series of corporate operation-related variables to remove the influence of other variables on the dependent variable. It is primarily comprised of Independent Director or Not (Indep), leverage ratios (LEV), and corporate age (AGE).

### 3.3. Empirical Model

This study provided an empirical framework that expands the concepts given in previous research [61,62,63]. On the basis of the selection of the aforementioned important indicators, an empirical evidence-testing regression model was developed. To test the proposed research hypothesis in this study, the following regression models are developed:

#### 3.3.1. Direct Effect Model

In this formula, control is the set of the control variable, the Constant is the intercept term, Ɛ represents the random disturbance term, and β represents the regression coefficient of each explanatory variable. Models 1, 2, and 3 are the regression models of corporate environmental governance between control variables and explained variables.
(2)$$\mathrm{CEG}=\varepsilon +\beta_{1}\mathrm{EU}+\beta_{2}\mathrm{Lev}+\beta_{3}\mathrm{FA}+\beta_{4}\mathrm{Indep}+\mathrm{Constant}$$
(3)$$\mathrm{CEG}=\varepsilon +\beta_{1}\mathrm{CR}+\beta_{2}\mathrm{Lev}+\beta_{3}\mathrm{FA}+\beta_{4}\mathrm{Indep}+\mathrm{Constant}$$
(4)$$\mathrm{CEG}=\varepsilon +\beta_{1}\mathrm{GI}+\beta_{2}\mathrm{Lev}+\beta_{3}\mathrm{FA}+\beta_{4}\mathrm{Indep}+\mathrm{Constant}$$

#### 3.3.2. Moderating Effect Model

To determine the moderating roles of environmental investment on the relationship between corporate environmental governance and sustainability development strategy. Model 4 added moderating variables on the basis of model 1 to test the impact of corporate environmental investment on the correlation between environmental uncertainty and environmental governance. On the basis of model 2, the interactive term between the carbon responsibility of listed companies and the corporate environmental investment is added to test the moderating effect of corporate environmental investment, that is, Hypothesis H5. Model 6 is used to highlight the interaction effects of environmental investment and green innovation on corporate environmental governance.
(5)$$\mathrm{CEG}=\varepsilon +\beta_{1}\mathrm{EU}+\beta_{2}\mathrm{Lev}+\beta_{3}\mathrm{FA}+\beta_{4}\mathrm{Indep}+\beta_{5}\mathrm{EU}*\mathrm{EI}+\beta_{6}\mathrm{EI}+\mathrm{Constant}$$
(6)$$\mathrm{CEG}=\varepsilon +\beta_{1}\mathrm{CR}+\beta_{2}\mathrm{Lev}+\beta_{3}\mathrm{FA}+\beta_{4}\mathrm{Indep}+\beta_{5}\mathrm{CR}*\mathrm{EI}+\beta_{6}\mathrm{CR}+\mathrm{Constant}$$
(7)$$\mathrm{CEG}=\varepsilon +\beta_{1}\mathrm{GI}+\beta_{2}\mathrm{Lev}+\beta_{3}\mathrm{FA}+\beta_{4}\mathrm{Indep}+\beta_{5}\mathrm{GI}*\mathrm{EI}+\beta_{6}\mathrm{GI}+\mathrm{Constant}$$

The control variables and intercept terms are similar to the direct effects model. The above-mentioned variables present the interactions between the sustainability development strategy and the moderators (corporate environmental investment), where the link between the product of the variables and corporate environmental investment is used as a regressor. 

## 4. Results and Discussion

### 4.1. Descriptive Statistics Analysis

Table 1 displays descriptive statistics for each variable. The table reveals that the average value of environmental governance (EG) is 0.493, the minimum value is 0 and the maximum value is 30, demonstrating that firms experience varying levels of environmental governance. The standard deviation of environmental investment is 2803, showing that the average level of environmental investment among listed companies is adequate. The standard deviation of environmental uncertainty (EU) is 1.557, with a maximum of 28,718 and a low of 7125, demonstrating that environmental uncertainty varies amongst companies. The minimum carbon responsibility (CR) value is 2398; the highest value is 18,582; the standard deviation is 3011. There are disparities amongst businesses in terms of their carbon obligation. The smallest value of green innovation is 0, the mean value is 6863, and the maximum value is 139. The considerable gap demonstrates that the outcomes achieved by diverse firms after progress in green innovation are vastly different and that input and output are not necessarily correlated. As a result, there are differences in enterprise-to-enterprise transition achievement.

### 4.2. Correlation Analysis

Table 2 is the regression table of the correlation between the main variables. The correlation coefficients of independent and dependent variables were statistically significant and positive. This indicates that in a linear relationship, corporate sustainability strategies and corporate environmental governance outcomes are positively and statistically significant. This implies that sustainable development strategies are beneficial to corporate environmental governance. 

### 4.3. Regression Analysis

#### 4.3.1. The Results of a Direct Relationship

This section discusses the direct relationship between environmental uncertainty, corporate carbon responsibility, green innovation, and corporate environmental governance. 

In order to investigate the relationship between the two in detail, Table 3 provides the findings of analysing Hypotheses 1, 2, and 3. First, the results of hypothesis 1 reveal that environmental uncertainty has a statistically significant positive relationship with corporate environmental governance at the 1% confidence level. This indicates that when environmental uncertainty increases, Chinese publicly traded companies’ environmental governance improves. In addition, the results of the second hypothesis reveal a statistically significant positive association between corporate carbon responsibility and company environmental governance, with a 95% confidence interval and a regression coefficient of 0.6157. As a result of carbon duty, corporate environmental governance will improve. Thirdly, there is a substantial positive association between green innovation and environmental governance. This shows that the expanding capacities of green innovation in corporations are good for environmental governance outcomes.

#### 4.3.2. The Results of Moderating Effect

This section analyses the moderating effect of corporate environmental investments on the relationship between corporate environmental governance and corporate sustainability development strategy.

The regression results for the moderating impact are shown in Table 4. At the 10% confidence level, the regression findings for hypothesis 4 indicate that the coefficient of the moderating influence of corporate environmental investment on the connection between environmental uncertainty and environmental governance is 0.248, which is statistically significant. This indicates that corporations invest more in environmental protection and that environmental uncertainty reduces the effectiveness of environmental governance. The model result for column (2) of Table 4 indicates that the moderating effect of environmental investment on corporate carbon responsibility and environmental governance is minor and not statistically significant. However, it is important to note that in the direct relationship, corporate carbon responsibility has a positive effect on environmental governance, and despite the fact that the moderating effect is not statistically significant, the regression coefficient is −0.0759, which is less than 0. This indicates that despite the fact that the association is statistically insignificant, it changes. Both regression coefficients are positive and significantly positively associated, supporting the conclusion that environmental investment moderates the relationship between green innovation and environmental governance. This indicates that the moderating effect of environmental investment increases the influence of green innovation and environmental governance.

### 4.4. Robustness

This study tests the robustness by replacing the measurement index of corporate environmental governance to ensure the reliability of the empirical results. The specific method is to measure corporate environmental management system certification by listed companies. The regression analysis is continued according to the model set out above. The specific operation results are shown in Table 5:

To determine the robustness of the moderating impact, this study does additional analyses. Consistent with the above information, the results of the primary models are all determined to be statistically significant, with only modest fluctuations at the significance level. 

In Figure 2 and Figure 3, this study utilised the method of Aiken and West [64] to graphically depict the moderating effects in question (1991). These graphs illustrate the effects of low (−1 standard deviation from the mean) and high (+1 standard deviation from the mean) environmental expenditure on corporate environmental governance. Consistent with Hypothesis 4, Figure 2 demonstrates that corporate environmental governance reaches its peak when both corporate environmental investment and environmental uncertainty are high. Figure 3 illustrates that when environmental investment is low, corporations have higher levels of environmental governance, a significant new discovery.

## 5. Conclusions

The impact of corporate sustainability initiatives on corporate environmental governance is analysed using data from Chinese listed businesses from 2016 to 2020, based on previous research. This study also investigates the moderating effect of corporate environmental investment on sustainable development strategies and corporate environmental governance and provides conclusions. Environmental uncertainty, corporate carbon duty, and green innovation all have a favourable effect on corporate environmental governance, according to the empirical findings. This is a promising outcome because it provides a firm answer to address environmental governance and long-term environmental degradation challenges.

In particular, corporate environmental governance outcomes are better the more environmental uncertainty it faces. This may be because, for a while, the corporate sustainability strategy has emphasised financial performance, such as the ability of the corporation to capitalise [48]. Additionally, China’s goal of being “carbon neutral” makes environmental governance for corporate much more difficult and urgent [12]. In light of this, it’s possible that growing environmental uncertainty is to blame for the results of corporate environmental governance during this period. Secondly, the empirical findings of this study demonstrate a favourable correlation between corporate environmental governance outcomes and carbon responsibility. This suggests that corporate improved environmental governance outcomes are supported by their decreased GHG emissions. According to earlier research, listed companies of China should prioritise environmental responsibility to acquire strategic environmental advantages and reap benefits for the environment and the economy [8]. Similar to [65], they assert that the carbon market has improved the efficacy of environmental protection. The study concludes that the development and achievement of corporate carbon responsibility targets represent a new stage in corporate environmental responsibility, while making a positive contribution to environmental governance. Additionally, this study’s findings support earlier research in that they show that green innovation has a favourable impact on corporate environmental governance outcomes [2,66]. This finding lends credence to the notion that corporations should pursue green innovation in order to enhance environmental governance in the run-up to achieving their long-term sustainability objectives.

As for the findings on the moderating effect of environmental investment, the empirical results of this study show that environmental investment has a significant moderating effect on corporate environmental uncertainty and green innovation, but not on corporate carbon responsibility. Specifically, greater environmental investment by corporate diminishes the impact of environmental uncertainty on environmental governance. This implies that when environmental investment rises, the effect of environmental uncertainty on corporate environmental governance reduces. New arguments for long-term corporate strategies to mitigate environmental issues are presented by this finding, which also offers new evidence and ideas for corporate environmental management practices. It also highlights the necessity of taking into account how investments in the environment will affect environmental uncertainty when developing corporate sustainability strategies. For green innovation, corporate environmental governance will be aided by increased corporate investment in corporate environmental protection practices. This is consistent with earlier research [7] that supports corporate innovation as a means of reducing environmental issues. The fact that greater environmental investment greatly enhances the effects of corporate green innovation on environmental governance also serves as compelling evidence for future corporate environmental governance. Green innovation is advantageous for corporate solutions to environmental issues in the future, and environmental investment will further this advantage.

In conclusion, the adoption of environmentally sustainable development strategies by Chinese companies has effectively addressed the issue of corporate environmental governance. Moreover, corporate environmental investment has successfully mitigated the influence of corporate sustainability policies on environmental governance challenges. The empirical results indicate that the environmental governance outcomes of Chinese enterprises are beneficial, but the moderating effect of corporate environmental investment is not a unique conclusion in the face of environmental legislation and stakeholder pressure. After appropriate analysis, businesses should pursue further sustainable development methods and environmental investments. 

## 6. Implications and Limitations

The following are the main implications arising from this study. 

Firstly, this study combines environmental economics theory and resource-based theory to examine the impact of corporate sustainable development strategy on corporate environmental governance. In the current context, the implementation of a sustainable development strategy by a Chinese corporation facilitates corporate environmental governance outcomes. This enriches the research on the resource-based theory of corporate and corporate environmental governance. In addition, this study explores the moderating effect of corporate environmental investment on sustainable development strategies and corporate environmental governance. The empirical results of this study realistically show that corporate environmental investment has a significant impact on environmental uncertainty and green innovation, thus this finding fills an existing research gap.

Furthermore, this study has contributed to business executives in implementing sustainable development strategies in practice to promote corporate environmental governance. Corporate executives should strengthen their knowledge of corporate environmental governance and find the best way to integrate it with their current corporate production operations. In addition, the right environmental investments give Chinese corporate an edge in sustainable development. Therefore, executives should focus their environmental investment to enhance their corporation’s core green competencies. Managers are paying more attention to environmental issues to satisfy environmental responsibility. Green innovation should be promoted at a strategic level and truly integrated into actual operations, with joint cooperation across borders.

In addition, in the face of a complex and changing international environment and China as an emerging market, this study provides a different understanding and perspective on other emerging markets. For other countries and markets, the empirical results of this thesis give evidence of an emerging market and a deeper understanding of the impact of corporate sustainability strategies and environmental governance issues, and environmental investments on other countries or regions in the future. 

Despite the fact that this study offers valuable information, there are several limitations because of the authors’ constraints. First of all, this study excludes all other emerging markets and only focuses on listed firms in the China region. Secondly, this study only considers factors at the corporate level; future research might take more levels of factors into account. Thirdly, since sustainable development is a long-term plan, the long-term effects of the unique time period of the COVID-19 pandemic may be taken into consideration. 

## Figures and Tables

**Figure 1 ijerph-20-04528-f001:**
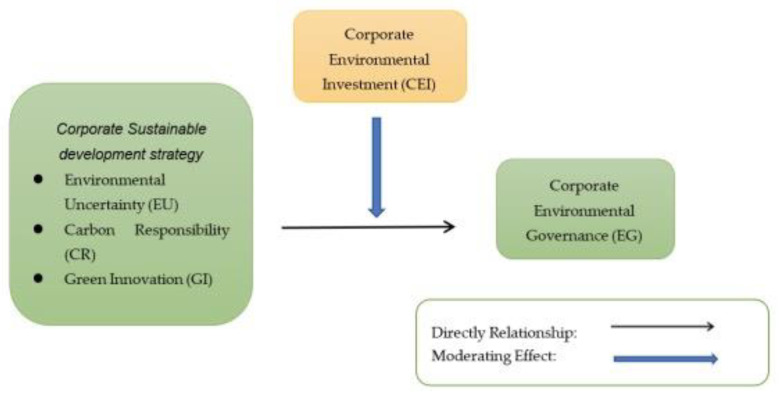
Research Framework.

**Figure 2 ijerph-20-04528-f002:**
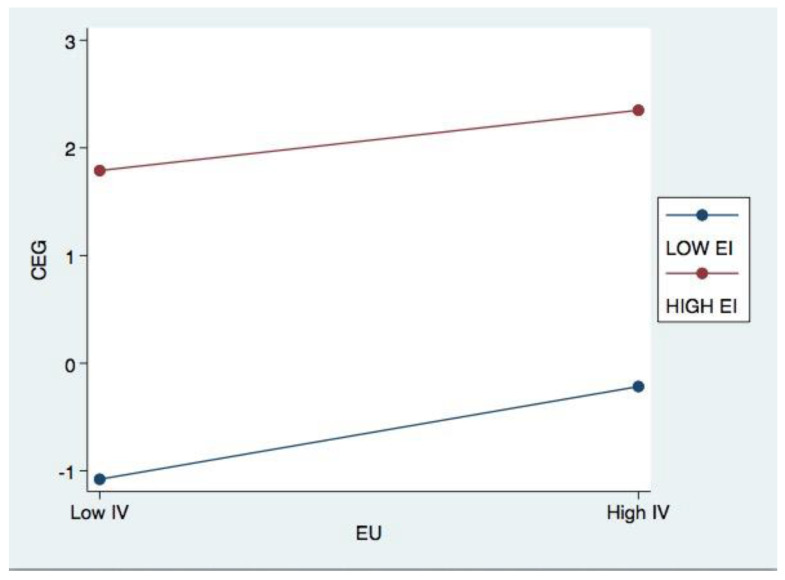
Effects of EI on the relationship between EU and CEG.

**Figure 3 ijerph-20-04528-f003:**
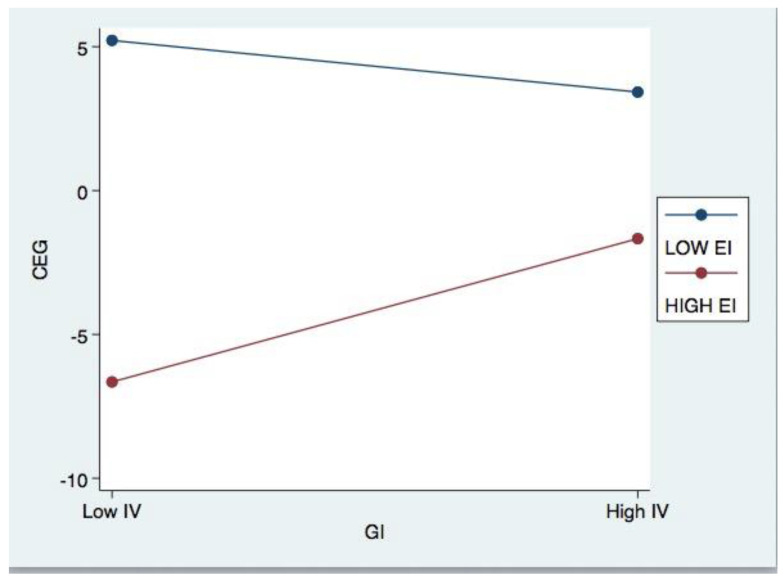
Effects of EI on the relationship between GI and CEG.

**Table 1 ijerph-20-04528-t001:** Descriptive Statistics.

Variable	N	Mean	SD	Min	Max
Environmental Governance	14,512	0.493	2.777	0	5
Environmental Investment	14,512	5.719	2.803	0	10.20
Environmental Uncertainty	14,512	21.197	1.557	7.125	28.718
Carbon Responsibility	14,512	10.559	3.011	2.398	18.582
Green Innovation	14,512	6.863	19.106	0	139

**Table 2 ijerph-20-04528-t002:** Correlation Analysis.

	Environmental Governance	Environmental Investment	Environmental Uncertainty	Carbon Responsibility	Green Innovation
Environmental Governance	1				
Environmental Investment	0.395 ***	1			
Environmental Uncertainty	0.110 ***	0.054 ***	1		
Carbon Responsibility	0.166 ***	−0.0140	−0.0240	1	
Green Innovation	0.048 ***	0.016 **	0.402 ***	−0.0240	1

*** *p* < 0.01, ** *p* < 0.05.

**Table 3 ijerph-20-04528-t003:** The results of a direct relationship.

	(1)	(2)	(3)
Variables	CEG	CEG	CEG
EU	0.289 ***		
	(13.22)		
CR		0.6157 *	
		(2.44)	
GI			0.0945 **
			(2.95)
Lev	0.0256	0.720	0.851 ***
	(0.16)	(0.46)	(6.25)
FirmAge	−0.259 **	−0.482	−0.161
	(−2.77)	(−1.71)	(−1.79)
Indep	−0.352	5.374 **	−0.654
	(−0.68)	(2.63)	(−1.32)
Constant	−4.636 ***	8.074	1.127 *
	(−8.55)	(1.17)	(2.31)
Observations	14,512	14,512	14,512
R-squared	0.0162	0.0906	0.0053

*t* statistics in parentheses; *** *p* < 0.01, ** *p* < 0.05, * *p* < 0.1.

**Table 4 ijerph-20-04528-t004:** The results of moderating effect.

Variables	Corporate Environmental Governance		
	(4)	(5)	(6)
EU	**0.248 *****		
	**(12.35)**		
CR		−0.0759	
		(−0.57)	
GI			**0.286 *****
			**(4.35)**
Lev	0.0640	4.251	0.909 ***
	(0.43)	(1.38)	(4.51)
FirmAge	−0.282 **	−1.679	−0.270 *
	(−3.29)	(−1.07)	(−2.17)
Indep	−0.254	−11.41	−0.695
	(−0.54)	(−1.26)	(−1.03)
Constant	−3.840 ***	9.166	0.653 ***
	(−7.70)	(1.46)	(9.66)
EI	−0.152 *	−0.114	−0.0818 **
	(−2.47)	(−0.26)	(−2.96)
**EU×EI**	**−0.155 ****		
	**(−2.58)**		
CR*EI		0.0000215 ***	
		(6.05)	
**GI×EI**			**0.0164 ****
			**(2.98)**
Observations	14,512	14,512	14,512
R-squared	0.1708	0.3702	0.0292

*t* statistics in parentheses; *** *p* < 0.01, ** *p* < 0.05, * *p* < 0.1.

**Table 5 ijerph-20-04528-t005:** Robustness results.

	(1)		(2)
	Y		Y
EU	0.0487 ***	GI	0.0545 ***
	(11.61)		(3.34)
EI	1.467 ***	EI	−0.0154 *
	(5.06)		(−2.27)
EU*EI	−0.0381 **	GI*EI	0.00308 *
	(−2.97)		(2.28)
Lev	0.0492	Lev	0.284 ***
	(1.59)		(5.53)
FirmAge	−0.0589 **	FirmAge	−0.0614
	(−3.29)		(−1.94)
Indep	−0.0486	Indep	−0.142
	(−0.49)		(−0.83)
_cons	−0.749 ***	_cons	0.263 *
	(−7.19)		(2.30)
*N*	14,512	*N*	15,323

*t* statistics in parentheses; * *p* < 0.05, ** *p* < 0.01, *** *p* < 0.001.

## Data Availability

The datasets used and/or analyzed during the current study are available from the corresponding author on reasonable request.

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
