# Peer review of "Corporate Sustainability Development Strategy and Corporate Environmental Governance—The Moderating Role of Corporate Environmental Investments"

_ijerph, 2023, doi:10.3390/ijerph20054528_

Round 1

Reviewer 1 Report

The topic is critical and the authors have the merit of approaching it at a scholarly level.

There are several typo errors along the whole manuscript that need to be addressed by the authors (lines 74-75; 90; 106; 163…).

[line 253]: As the manuscript discuss the environment (in the meaning of nature) it is not prudent to name the variable describing the external features of the enterprise as “environmental” because it creates confusion. Please revise it and rename it.

Section 3.3 should present more details about the empirical model.

We also strongly encourage the authors to review recent studies approaching the Chinese case related to different factors influencing environmental quality and its features, such as:

https://doi.org/10.1108/SAMPJ-12-2019-0448

https://doi.org/10.1016/j.pacfin.2021.101596

https://doi.org/10.3389/fenvs.2023.1120970

https://doi.org/10.3390/ijerph18010150

Author Response

Daer reviewer,

Thank you for your valuable advice! Based on your comments, this article has been improved in terms of formatting and other details. In addition, improvements have been made to the construction of the paper's model and other aspects. Thank you very much for your time and review!

Best regards!

Reviewer 2 Report

The paper is interesting however I suggest some corrections to improve its quality:

1. The paper needs editing corrections, e.g. lines: 5, 11, 12,32,74-75,90,106,163

2. Abstract should be rewritten. I do not like the sentence: "And try to analysis the relationship between...". Please do not start the sentence with "and", please replace "to analysis" with "to analyse" etc.

3. All hypotheses should be written in bold or italics, not both of them

4. Fig. 1 needs improvement. Please exchange "directly relationship" for "direct relationship"

5. The title of the Tab. 1 should be "Descriptive Statistics"

5. Please do not start point 4.3.2. with the Table, add before the table some text

6. You use different fonts in tables, please use one type of font

7. We usually do not interpret the value of SD. It is presented in the table so it is not necessary to double it.

8. In H1 and H2 do you mean a statistically significant relationship? If yes please add the word 'statistically'

9. In H4, H5, and H6 "a" is missing

10. The title "Descriptive Statistical Analysis" is incorrect. You can use e.g. "Analysis of Descriptive Statistics" or "Descriptive Statistics Analysis"

Author Response

Dear Reviewer,

Thank you for your decision and valuable comments on my manuscript. We have made the following changes based on your suggestions. Firstly, this study has refined the details in accordance with the review comments in terms of formatting, font and others. And the details of the use of vocabulary such as the missing "a" and "Statistics" have been improved. Add some text in section 4.3.2.

Thank you very much for your valuable advice! 

Reviewer 3 Report

The paper addresses an interesting topic of research.  

The abstract is a little hard to understand. In this section of the paper, we must have information about the purpose of the paper, the methodology used, and the main findings of the study. Together with the title, this is the most read part of the paper. If this is not properly written then most likely nobody will read the full paper.

In this part, you have a sentence like “The Chinese government is attempting to guide and improve these environmental problems and therefore plans to improve the ecological environment by the end of 2020”, we are in 2023, maybe you can provide some updated information or you can present as a past action, not a future one.  I think that you need to revise the entire abstract so it will better express your work.

In the introduction part, at lines 34 and 35, you repeat the word “therefore”, try to reformulate it. At lines 52 and 53 you repeat “however”.

In lines, 74 and 75 you have different heights for the text, if you what to highlight that part you can use italics or bold. The same difference in text height can be seen in some citations.

The hypotheses are well formulated, but there is also a text formatting error, H 3 is written in italics while the others are bold, the same in the case of H 6, you need to have a certain uniformity.

Try not to use acronyms without explaining them (like CSMAR and so on).

You do not have a conclusion section. Even if you have a discussion part, and a limitation part, I think that you can add a small conclusion where you can highlight the main findings of your study. When reading a scientific article after the title, you read the abstract, and then for a little more information you read the conclusions. So these parts are extremely important to the success of a paper. I recommend revising these sections so you can better present your work.

Also please add some info about what can international readers learn from your experience. This is a case study, explain briefly what are the main aspects that can be a lesson for the international context. 

The bibliography is not uniformly presented. You need to choose a reference style and be consistent, and apply it throughout the entire paper. The references are written in different styles, you need to revise this part.

In general, I consider that you need to proofread again the entire paper, the English language is not the best, a lot of sentences are hard to understand.

The general impression is that you work very much and you have an interesting topic of research but it is not very well presented. You need to revised substantial your paper so all the data can be more easily presented, to have a certain flow of information and you need to think that not all the readers of your paper are familiar with this subject so you need to offer enough data that they can understand what you have done.

Good luck with your future research.             

Author Response

Dear reviewer,

Thank you very much for your valuable comments. We really appreciate your reviewing our manuscript. We have revised the manuscript accordingly. Firstly, we have improved the abstract section based on your comments, which is an addition to the methodology and a clarification of the purpose. In addition, this study improved repeat words in your review, that is however and therefore. Moreover, the format problems and acronyms have been refined and adjusted. Meanwhile, a conclusion section and suggestions for perspectives for international readers have been added to this paper. 

Thank you very much for your time and advice, and I wish you good health and all the best for your future research!
